# Role of Polymers in Microfluidic Devices

**DOI:** 10.3390/polym14235132

**Published:** 2022-11-25

**Authors:** Laila A. Damiati, Marwa El-Yaagoubi, Safa A. Damiati, Rimantas Kodzius, Farshid Sefat, Samar Damiati

**Affiliations:** 1Department of Biology, Collage of Science, University of Jeddah, Jeddah 23890, Saudi Arabia; 2Department of Pure and Applied Chemistry, University of Strathclyde, 295 Cathedral Street, Glasgow G1 1XL, UK; 3Department of Pharmaceutics, Faculty of Pharmacy, King Abdulaziz University, Jeddah 21589, Saudi Arabia; 4Faculty of Medicine, Ludwig Maximilian University of Munich (LMU), 80539 Munich, Germany; 5Faculty of Medicine, Vilnius University, 03101 Vilnius, Lithuania; 6Interdisciplinary Research Centre in Polymer Science & Technology (Polymer IRC), University of Bradford, Bradford BD7 1DP, UK; 7Department of Biomedical and Electronics Engineering, School of Engineering, University of Bradford, Bradford, BD7 1DP, UK; 8Department of Chemistry, College of Sciences, University of Sharjah, Sharjah 27272, United Arab Emirates

**Keywords:** polymers, microfluidics, lab-on-chip, biomedical engineering, drug carrier, artificial cell, 3D bioprinting

## Abstract

Polymers are sustainable and renewable materials that are in high demand due to their excellent properties. Natural and synthetic polymers with high flexibility, good biocompatibility, good degradation rate, and stiffness are widely used for various applications, such as tissue engineering, drug delivery, and microfluidic chip fabrication. Indeed, recent advances in microfluidic technology allow the fabrication of polymeric matrix to construct microfluidic scaffolds for tissue engineering and to set up a well-controlled microenvironment for manipulating fluids and particles. In this review, polymers as materials for the fabrication of microfluidic chips have been highlighted. Successful models exploiting polymers in microfluidic devices to generate uniform particles as drug vehicles or artificial cells have been also discussed. Additionally, using polymers as bioink for 3D printing or as a matrix to functionalize the sensing surface in microfluidic devices has also been mentioned. The rapid progress made in the combination of polymers and microfluidics presents a low-cost, reproducible, and scalable approach for a promising future in the manufacturing of biomimetic scaffolds for tissue engineering.

## 1. Introduction

Polymeric biomaterials have been used to provide artificial matrices that can mimic the biological cell. This requires appropriate biophysical and biochemical properties, such as certain topography, stiffness, signaling, and growth factors [1]. Polymers are commonly used in tissue engineering scaffolds and wound dressing due to their ability to enhance cellular regeneration. Further, drugs are encapsulated in the polymeric particles to generate drug vehicles that can improve drug uptake into the disease sites and bioavailability. However, the physical and self-assembled properties of polymers, such as charges, composition, biodegradation, shape, size, and surface chemistry, play a dominant role in determining polymer behavior within biological environments [2]. Further, these interactions are directed by the physicochemical properties of the polymers in micro or nanostructures. The developed polymeric models are able to navigate the body, infect and transform cells, or repair damaged cells. The incorporation of cells into polymeric matrix can be performed by cell implantation into readily prepared polymer matrix. This strategy has a significant drawback, namely the lack of good integration between cells and polymer matrix. Another alternative strategy relies on the fabrication the polymer matrix with encapsulated cells, which allows development of complex cellular microenvironments. New techniques, such as microfluidics, 3D printing, and electrospinning, enable direct cell integration into the matrix to mimic the matrix of desired tissue [1].

Microfluidics technology, also known as lab-on-chip technology, has been used as a platform for biomedical engineering applications [3]. Generally, a microfluidic chip is network of microchannels incorporated into the microenvironment by several holes throughout the chip. Microfluidics allow the integration of biological and chemical processes on a single platform. These microdevices allow controlling of the flow behavior of small volumes of fluids in micro-chambers in the range of tens to hundreds of micrometers. Microfluidics are widely used to synthesize polymeric particles for various applications involving drug carrier vehicles, as well as bioarchitecture models mimicking cell-like structures or extracellular matrix (ECM). Furthermore, recent studies have demonstrated the possibility of using microfluidic chips as an artificial cell chassis. Depending on the application, glass or polymer can be used to manufacture microfluidic devices and several parameters should be taken into consideration in the fabrication of a microfluidic chip, such as the compatibility of constructed materials with various solvents and channel geometries [4,5,6,7]. Polymer-based chips are usually selected due to their cost efficiency, suitable optical transparency, elasticity, and appropriate mechanical and chemical properties [8]. Several polymers, such as polydimethylsiloxane (PDMS) and poly(methyl methacrylate) (PMMA), are employed to fabricate microfluidic devices [3]. However, polymers have some limitations regarding their properties, including operation temperature range limitations, higher autofluorescence, and the limited availability of surface modification techniques [9]. The fabrication of polymer microfluidic devices is relatively simple, and hazardous etching reagents are not needed to create the polymer microstructure [10].

It is common to use natural polymers, such as polysaccharides and bacterial polyesters, to generate polymer-based therapeutics, while it is common to use synthetic polymers as building blocks for microfluidic devices. Figure 1 shows different applications of utilizing polymers and microfluidics. Biodegradable and bioreducible polymers that are used for polymeric drug/gene delivery systems are rapidly emerging in pharmaceutical fields. Combining therapeutic agents with polymers can improve their safety and efficacy by controlling the rate, time, and preferentially delivers the therapeutic agents to the target site in the body [11]. Combining microfluidic devices and polymers presents unique advantages for the development of efficient carriers of a wide range of drugs and genetic materials (e.g., polymersomes). Microfluidic technology enables the production of highly stable, uniform, monodispersed particles with higher encapsulation efficiency [5]. Furthermore, many studies showed the possibility of using polymer-based bioinks in 3D printing for applications in tissue engineering and regenerative medicine. There are several natural (e.g., alginate, collagen, agarose) and synthetic (e.g., Pluronic and poly(ethylene glycol) (PEG)) polymeric biomaterials that are used as bioinks for 3D printing based on their ability to support cell growth, mechanical properties, and printability. Combining of cells, biomedical polymers and biosignals is the basic requirement to develop 3D tissues or organ structures [12]. The fabrication of vessel-like microfluidic channels is an example of organ fabrication and thick tissue. Besides supporting the mechanical integrity, the printed 3D microfluidic network enables fluid transport. The microfluidic architecture allows media transport, including nutrients, oxygen, water, and removal of the waste in the same manner [13]. Recent developments in droplet microfluidics allowed the creation of versatile vesicles with a structure that resembles the biological membrane. These artificial cell-like structures with well-defined size enable the implementation of various biological reactions within a compartment separated by a membrane that mimics a natural cell membrane [3]. In this perspective, this review deals with polymers used to either fabricate microfluidic devices or create functional particle/matrix models using polymers and microfluidic chips. The review first provides an overview of the different types of polymers. Then, it highlights some of the recent advances in the design of microfluidics and polymers for various biomedical engineering applications, including drug delivery, 3D bioprinting, and artificial cell-like structures.

## 2. Polymers Used in Microfluidic Devices

A single polymer unit may be composed of hundreds or millions of monomers. They have one of the four basic polymer structures: linear, branched, cross-linked, or networked (Figure 2). The two types of polymers are natural and synthetic. Natural polymers can be extracted from biological systems such as plants, algae, microorganisms, and animals, which have a similar ECM structure to native tissues. Synthetic polymers are similar to natural polymers, but they are much cheaper, can be produced at large scale, and have long shelf life compared with natural polymers. As such, generally, they present good cellular attachment, which improves the cellular behaviors and prevents immunological reactions [1,14].

Choosing the right material is the first and most critical step in designing a successful microfluidic device. A wide range of constraints and requirements dictates the selection of the material for a specific component. The design of the device, the compatibility of the material with the chemicals, as well as the applied temperature and pressure are crucial considerations in material selection. Additionally, the final application of the device is an essential consideration. For example, devices intended for in vivo applications in tissue engineering must be nontoxic, exhibit a slow and predictable degradation rate, have nontoxic and safe degradation products, and potentially capable of mimicking certain physical and chemical properties of the native ECM or of supporting other agents that play such roles. The architecture of the device can also influence the choice of materials. For example, in devices that contain microfluidic systems, the materials have to be mechanically robust but have a controlled degree of flexibility [15,16]. The material also has to be compatible with microfabrication techniques, easily processed in mild conditions, and cheap to manufacture, among others [17]. Polymers are classified into two major groups: biodegradable and biostable polymers. These two types of polymers are commonly used as scaffolds or bioactive coatings in biomedical applications [8,15]. The next section focuses on these two classes of polymers for the manufacture of microfluidic devices and their biomedical applications.

### 2.1. Biodegradable Polymers

Sustainable polymers from various renewable resources can be directly obtained from biomass (proteins and polysaccharides), or through chemical modifications of natural polymers [18]. However, there are many sources of natural and synthetic biodegradable polymers. Natural polymers are derived from natural raw materials and available in large quantities while synthetic polymers are synthesized by the chemical polymerization of bio-monomers.

#### 2.1.1. Natural Biodegradable Polymers

In this section, we distinguish between natural polymers, which are produced outside the human body (xenobiotic polymers), and proteins, which are native to the human body, such as the ECM proteins. The use of natural biopolymers in microfluidics provides many advantages, such as surface chemistry biocompatibility and having the same mechanical properties of the native proteins of interest [16,19].

Natural polymers, such as chitosan, alginate, and gelatin, are also biologically derived and biodegradable polymers. They are used in the manufacturing of biodevices that are intended to interact with the biological systems of the human body. The crosslinking ability of these natural polymers, which is induced by physical and chemical stimuli, makes them ideal for the preparation of microgels for microfluidic devices. The two natural biopolymers, alginate and gelatin, were used as substrates to make two types of hydrogel-based microfluidics. Subsequently, the fabricated hydrogel microchannels can be used as platforms to provide 3D cell culture environments for mammalian cells: fibroblasts and vascular endothelial cells. The developed enclosed microchannel models are simple and reproducible and do not require complicated operations [16,19].

One class of natural biomaterials that is a good candidate for microfluidic devices is silk fibroin (SF) [20]. SF protein, originally found in the silkworm *Bombyx mori*, is a high-molecular-weight protein that primarily consists of hydrophobic residues. This protein is approved by Food and Drug Administration (FDA) for many medical applications, such as drug delivery and tissue engineering. SF can be easily processed to form hydrogels, films, and nanofiber mats under mild conditions [21]. In recent years, SF has also been used to fabricate microfluidic devices due to its excellent biocompatibility, robust mechanical properties, and slow proteolytic degradation rate [16,20]. The solubility and mechanical properties of SF materials are linked to its secondary structure. Whereas self-assembled β-sheet structures are responsible for the mechanical stability and water insolubility of SF, the amorphous regions, including random coil, α helix, and β turn structures, contribute to the elasticity and solubility of the biomaterial. Thus, SF-based microfluidic fabrication strategies allow the rapid and scalable production of devices without the need for harsh processing conditions that require cytotoxic reagents. Mao et al. used SF and chitosan to construct porous SF–CS scaffolds with predefined microfluidic channels. The generated model showed structural properties suitable for seeding and growth of hepatic cells. Mass transport and uniform cell distribution within the 3D scaffold were successfully achieved [19].

In addition to all the above-mentioned advantages of natural polymers, the inclusion of natural ECM proteins into microfluidic devices allows the reproduction native cell–biomaterial interactions in vitro [22]. The use of ECM proteins is crucial in controlling cell function overall via other physicochemical mechanisms such as specific cell binding domain sequences. Proteins, such as fibronectin, vitronectin, and collagen I, contain the amino acid sequence of arginine–glycine–aspartic acid (RGD), which supports the adhesion of cells and to control stem cell differentiation. For example, Arik et al. reported the fabrication of a collagen-I-based membrane incorporated in an organ-on-chip device [23]. The membrane demonstrated permeability, as well as the adhesion of both endothelial and epithelial cells. Moreover, they characterized the degradation and remodeling of the basement membrane by a protease. Natural proteins offer an environment that more closely mimics that of the body and more realistically mimics the cell–ECM interactions, which are crucial for tissue engineering. However, these biomaterials have a complex structural composition that prevents complete control over their composition and other factors, such as molecular weight, immune response, degradation, and mechanical properties. As an alternative, scientists have focused on the development and use of synthetic polymers, which have more tunable properties [22].

#### 2.1.2. Synthetic Biodegradable Polymers

Synthetic polymers were proposed as ideal candidates for the fabrication of biodegradable microstructures, including microfluidic biomaterials [16,24]. Poly(glycolic acid) (PGA), poly(lactic acid) (PLA), and their copolymer poly(lactic acid-co-glycolic acid) (PLGA), belong to the linear aliphatic polyesters family [25] (Figure 3). This polymer family is one of the most widely used in tissue engineering and drug delivery [25,26]. These polymers have several advantages, such as low cost, ease of processing, and well-characterized biological behavior. These polymers (PLA, PGA, and PLGAs) are among the few synthetic polymers approved by the U.S. FDA for certain human clinical applications [26]. These polymers degrade through the hydrolysis of the ester bonds [27]. Although PGA and PLA belong to the same family, they also display distinct properties. For instance, because of its very hydrophilic nature, PGA rapidly degrades in aqueous solutions. However, PGA and PLA show the same behavior in vivo: they lose mechanical integrity in a period between two and four weeks [28]. Conversely, PLA contains a methyl group, which renders the chains more hydrophobic and hence reduces the affinity to water, and displays a slower hydrolysis rate (months to years) [28]. This class of biodegradable polymers is suitable for microfluidics because of the wide range of tunable properties [27]. They can be modulated by adjusting the lactide-to-glycolide ratio.

The physical properties of the copolymer PLGA are defined by the properties of both pure PGA and PLA [25]. The presence of PLA makes it more hydrophobic than PGA. Hence, lactide-rich PLGA copolymers are less hydrophilic and more slowly degrade. Additionally, PLA exhibits relatively a high glass transition temperature (Tg = 50–80 °C) and melting point (Tm = 173–178 °C). PLGA blends of various copolymer ratios exhibit a reduced phase transition temperature (Tg, PLGA75/25 = 54 °C) and melting point (Tm, PLGA75/25 = 80 °C) [29]. Poly(caprolactone) (PCL) is another example of an aliphatic polyester used in microfluidics. PCL demonstrates advantageous properties for replica molding strategies, such as a low melting point (Tm = 57 °C) and low glass-transition temperature (Tg = −62 °C) [30]. PCL can be degraded by micro-organisms as well as by the hydrolysis of its ester linkage in physiological conditions [31]. However, PCL materials have a substantially slower biodegradation rate than PLA and PGA, making it suitable for the use in long-term implantable systems. Biodegradable cell-support scaffolds play an important role in the growth of engineered tissue and the delivery of biologically active agents. Therefore, the concept of biodegradable microfluidic devices formed by various biodegradable polymers has attracted considerable research attention. For example, microstructured PLGA films were used to construct a high-resolution and high-precision 3D device. The developed device allows diffusion distance reduction in cell-seeded scaffolds with convective transport [32]. PLA microchannels have been widely generated by 3D printing. Kadimisetty et al. developed a microfluidic immunoarray using PLA and a 3D printer. The fabricated device was low cost and could sensitively detect prostate cancer biomarker proteins [33].

Poly(1,3-diamino-2-hydroxypropane-co-polyol sebacate) (APS) is another biodegradable elastomeric polymer used to construct microfluidic scaffolds. The simple microchannel network design exhibited a very low degradation rate while retaining the elastomeric properties required for tissue scaffold applications [24].

### 2.2. Biostable Polymers

PDMS is a mineral–organic polymer structurally composed of silane-oxygen backbones covered with alkyl groups. Depending on the size of the monomer chain, non-cross-linked PDMS may be almost liquid (low amount of n monomer) or semi-solid (high amount of n monomer) [34]. The high level of viscoelasticity displayed by the polymer chain is due to the siloxane bonds in the polymer structure. After cross-linking with a curing agent, PDMS becomes a hydrophobic elastomer [34]. One of the main reasons for the success of PDMS in microfluidics is the ease of PDMS device fabrication, which also allows mass production. Among many other methods, PDMS microchips can be fabricated through microscale molding processes [35]. For example, a silicon wafer with patterns can be used as a mold master. Prepolymer PDMS is poured into the mold master. Then, cured PDMS is peeled off from the master to be pasted on a flat plate, i.e., PMMA, glass, etc. [34]. The flat support should be drilled in advance to provide access ports for the introduction of reagents and samples. PDMS can precisely replicate structures down to the submicron size [36]. Due to the favorable optical properties of PDMS (almost no absorbance in the visible wavelength range), fluorescent dyes are widely used for the detection and quantification of molecules in most biochemical analyses. In addition, PDMS is transparent, biocompatible, nontoxic, and displays high gas permittivity, so has been traditionally used as a biomaterial in catheters, insulation for pacemakers, and ear and nose implants [10].

The combination of its elastic properties, easy processability, and the other properties mentioned above make PDMS an ideal candidate for use in microfluidic devices for biomedical and cell applications.

Many studies have been performed to further examine the compatibility of PDMS with both microfluidic technology and biomedical applications [37]. In terms of microfluidic technology, the effects of the structure and surface of PDMS in widely used microfluidics methods, such as spin coating and chemical immersion, on different liquid chemicals have been studied. Successful spin-coating of PDMS depends on the crosslinking ratio; increased amounts of crosslinker agent in the formulation decrease film thickness. Additionally, whereas chemical immersion (solvents such as alcohol, toluene, acetone, etc.) does not result in major changes in the surface hydrophilicity of PDMS, macrotexture distortion and destructions are observed with strong acids (hydrofluoric, nitric, sulfuric, and hydrofluoric acids) and bases (potassium hydroxide). For biomedical applications, the effect of oxygen plasma and sterilization and the exposure to tissue culture media was also explored. Oxygen plasma exposure increases PDMS surface hydrophilicity, whereas a following exposure to air leads to hydrophobic recovery. UV and alcohol sterilization do not affect the PDMS surface microtexture, element concentration, hydrophilicity, or mechanical properties. Finally, immersion in tissue culture media increases the surface concentration of oxygen relative to silicon [38].

Despite all these advantages, the use of PDMS is limited due to challenges encountered in microfluidics. For example, incomplete curing of PDMS leaves uncrosslinked oligomers within the material, which can leach out and contaminate the culture medium. Other problems, such as incompatibility with some organic solvents, water evaporation, channel deformation, and adsorption of biomolecules onto channel walls, present severe limitations to the use of PDMS for microfluidics applications [39].

#### Thermoplastics

Thermoplastics are plastic polymer materials that have emerged as a commercially viable material. Their use has recently increased, being widely applied to fabricate microfluidics platforms for biomedical applications. The most commonly used thermoplastics are PMMA, polycarbonate (PC), polystyrene (PS), polyvinyl chloride (PVC), Cyclo-olefin-copolymer (COC), and Cyclo-olefinpolymer (COP) [39,40] (Figure 4).

Because of their linear structure, their thermoplastic rigidity resists temperature and pressure changes. The properties of the most common thermoplastics used for chips fabrication are summarized in Table 1. Thermoplastic-based materials have good physical and chemical characteristics, such as high chemical and mechanical stability; low water-absorption capacity; acid/base resistivity; and are suitable for mass production at low cost. In term of fabrication, thermoplastics can be softened after exposure to heat at their transition temperature (Tg), making them processable around this temperature. During cooling, the softened polymer hardens, and it takes the shape of the container or mold, without any chemical change. They can be reshaped multiple times by reheating, which is important for the molding and microfluidics fabrication process [41].

One of the first properties to consider in cell biology is biocompatibility. According to Table 1, most of the thermoplastics are biocompatible. However, for long-term applications, some of the materials can be problematic. For example, polycarbonates can be experience surface erosion during in vivo applications. In addition, bisphenol A (BPA), which is hazardous in food contact situations, might be released during hydrolysis.

PVC can release toxic gases during manufacturing, and nylon is a heat-sensitive material. Resistance to solvents is also a main criterion that must be considered for microdevice fabrication and biomedical applications (sterility). PS is widely used in molecular and cell biology studies due to its biocompatibility and its high resistivity to alcohols, polar solvents, and alkalis [50]. PMMA is affected by ethanol, isopropyl alcohol, acetone, and other important solvents used in microfabrication and sterilization [51]. When working with cell cultures, low water absorption is beneficial because the cells consume more oxygen from water, which can be limited by the absorption of water onto the polymer surface.

The optical properties of the selected material (e.g., transparency and autofluorescence) are crucial. Consequently, PMMA, polyethylene terephthalate (PET), and polypropylene (PP) are less suitable for applications that require further reactions inside the microfluidic devices under a microscope. Additionally, PC displays high autofluorescence, so PC is difficult to use when working with fluorescently labelled cells or materials. In contrast, PS has high transparency, and the surface of PS is suitable for long-term cell studies [41]. Table 2 highlights some studies that used polymers as a chassis or to functionalize sensing surface.

## 3. Polymers as Drug Carriers

Generally, a drug is any bioactive molecule, including medicine, small molecules, and proteins, e.g., growth factors and nucleic acids [63]. Different polymers have been used in drug delivery approaches: i. a drug can be directly incorporated onto scaffolds throughout the casting process [64], ii. bulk hydrogels [65,66], iii. drug reversibly and covalently conjugated to the matrix [67], iv. micro- or nanodrug particles spread on the surface [68,69,70]. However, all of these methods have advantages and disadvantages in terms of drug stability [63]. When manufacturing new drug delivery system, different factors should be taken into consideration for instance cost, efficacy, and properties differences. Advances in manufacturing techniques may produce more complex drug carrier designs to allow specific drug release targeted to a particular disease [71]. Using a microfluidic platform approach can allow generation of drug carriers that can meet the sophisticated requirements of biomedical applications [72] (Figure 5).

Drug delivery devices have potential to be used for various clinical applications, such as tissue regeneration, diabetes, oncology, and infectious diseases. Moradikhah et al. used a cross-junction microfluidic device to prepare alendronate-loaded chitosan nanoparticles. They showed that this system substantially enhanced the osteogenic differentiation of human adipose MSCs, so can be a suitable component of bone tissue engineering scaffolds [73]. Mora-Boza et al. illustrated that their fabricated hMSC-laden microcarriers based on in situ ionotropic gelation of water-soluble chitosan in a microfluidic device using antioxidant glycerlphytate and tripolyphosphate maintained cell viability over time and increased the secretion of paracrine factor [74]. An example of oral delivery drug was examined by Jaradat et al.; insulin was encapsulated into various PLGA nanoparticles prepared by the microfluidic technique. They found that the mucopenetrating heparin sulfate-conjugated PLGA nanoparticles enhance insulin permeability in a triple-cultured intestinal model compared with unmodified and free insulin nanoparticles [75]. Another model developed by Damiati et al. used PLGA to generate indomethacin-loaded PLGA microparticles employing a 3D flow-focusing microfluidic chip. This model not only successfully incorporates indomethacin, which is a poorly water-soluble drug and nonsteroidal anti-inflammatory drug, but the authors also developed an artificial neural network as in silico tool to predict size microparticles [76,77].

An example of using polymers in drug delivery in cancer is biodegradable polymeric nanocapsules. Oxaliplatin, irinotecan, and 5-fluorouracil chemotherapy drugs were encapsulated and carried on a coaxial glass capillary microfluidic device, which the potential for targeting tumors as the drug release could be controlled [78]. Hong et al. reported that the synthesized amphiphilic tri-chain tricarballylic acid-poly (ε-caprolactone)-methoxypolyethylene glycol (Tri-CL-mPEG) and enzyme-targeted tetra-chain pentaerythritol-poly (ε-caprolactone)-polypeptide (PET-CL-P) using microfluidics continuous granulation technology improved the bioavailability and antitumor effects of curcumin in a mouse model [79]. A recent review by Salari et al. provides a comprehensive assessment of studies in the field of polymer-based drug delivery for anti-cancer therapy. In their study, 71 papers were investigated, and they conclude that the polymeric nanoparticles have influential roles in cancer treatment comparing to the conventional chemotherapy. Polymeric nanoparticles were able to reduce the cytotoxicity following chemotherapy drug administration, enhance therapeutic agents solubility, and inhibit tumor growth rate [80].

As bacterial infections are posing a major threat to human health, in addition to increasing antibiotic resistance, new methods for bacterial detection are necessary to reduce disease spread. Recently, advances in antibiotic treatment have focused on the targeted delivery of antibiotics, as well as antibiotics alternatives, such as antimicrobial polymers, peptides, nucleic acids, and bacteriophages [81]. Borro et al. reported that by using polymyxin B-aliginate-Ca^2+^ microgels prepared by 3D printing, the microfluidic mixer affected the charge contrast and composition of the microgel formation and the interaction with bacteria-mimicking liposomes at different ionic strengths [82]. Additionally, a P-based nanoparticles delivery system was used as therapy against bacterial biofilm infections. Huang et al. used PLGA-based nanoformations combined with carbon quantum dots (CQDs) using a microfluidic flow-focusing pattern to load different types of antibiotics, e.g., azithromycin and tobramycin. They found that the azithromycin-loaded CQD–PLGA hybrid nanoparticles showed synergistic chemo-photothermally antibiofilm effects against *Pseudomonas aeruginosa* [83]. Norries et al. illustrated that the hydrogel developed from the poly(2-hydroxyethyl methacrylate) (PHEMA) and coated with ciprofoxacin antibiotic reduced the biofilm production of *Pseudomonas aeruginosa* [84].

## 4. Polymers as Bioink for 3D Printing

Three-dimensional (3D) printing is a development technique that has been used during the last decade to produce microfluidic devices. It has many advantages such as low cost, enabling the easy design of complex 3D structures and rapid prototyping. However, 3D printing has some limitations regarding the size of the microchannels and some final steps that are related to the laborious fabrication [85].

Different methods can be used to produce printed porous materials: i. curing a porous monolithic polymer sheet into the chosen pattern with photolithography, ii. screen-printing silica gel particles with gypsum, and iii. dispensing silica gel particles with polyvinyl acetate binder using a 3D printer. All three approaches can be successfully used in microfluidics [86].

Hydrophilic and hydrophobic polymers can be used to generate 3D-printed microfluidic droplets to prevent water-in-oil or oil-in-water droplets from sticking to the interior device surfaces. Warr et al. investigated two different approaches to avoid this issue: First, different resins were tested to evaluate their suitability for droplet formation and material properties. They found that the hexanediol diacrylate/lauryl acrylate resin forms the best hydrophobic solid polymer that prevents aqueous droplets from attaching to the device wall. Second, they formed a fully 3D microfluidic annular channel-in-channel geometry that forms droplets that do not contact channel walls. As such, this geometrical approach can be used with hydrophilic reins [87].

Distler et al. found that 3D-printed hydrogel is more electroactive and cytocompatible and enhances cell adhesion and proliferation compared with a 2D flat hydrogel. This kind of hydrogel formulation has shown promise in in vitro studies, cell therapy, and assisted tissue engineering electrical stimulation [88]. Wright et al. used a hydrogel composed of calcium crosslinked alginate (polypyrrole–alginate composite) as bioink for tissue engineering. They found that PC12 neural cells adhere and proliferate slightly better than alginate scaffold alone [89]. A compensation between metallic and polymer materials was used to fabricate a novel complex 3D structure. A soft polymer was cast and cured into a 3D-printed thin-shelled metallic mold, followed by metallic mold etching in an acidic solvent, which did not affect the soft polymer. This approach provided various polymeric complex structures [90].

At present, organ failure is a worldwide issue, and allograft organ transplantations are seriously limited due to donor organ shortages, immune rejection, and ethical conflicts, so finding an alternative solution is crucial [91,92,93]. Several polymers have been used for bioartificial organ manufacturing with different types of cells, e.g., stem cells, various growth factors, and vascular and neural networks. However, 3D bioprinting technology is a challenging engineering approach. Cooperation is required between different fields, such as biomaterials, biology, medicine, physics, chemistry, bioinformatics, and engineering, to fulfill all the requirements from the molecular to organ levels. Further, 3D bioprinting of polymers needs to meet several basic requirements to be applied in clinical applications. These requirements include biocompatibility, biostability, good mechanical properties, bioprintability, biodegradability, suturable with host vascular and nerves, permeability for nutrients and gases, and sterilizability [93].

## 5. Polymers as Artificial Cells or Organs

Numerous researchers have been trying to reduce the gap between the structures that can be designed and produced in the laboratory and those found in biology. Biological cells provide multiple functions, such as synthesizing proteins and lipids, storing genetic materials, storing and harvesting energy, etc. [92,94] (Figure 6). Additionally, homogeneous cells organize into specific tissues, whereas heterogeneous cells aggregate into an organ with specific physiological functions [93]. As such, creating an artificial cell or organ that has the same compartmentalized, multifunctional architecture is a challenging task. Two fundamental approaches have been considered for artificial cell constriction: top-down and bottom-up approaches. The top-down approach starts from living organisms, moving down the genome to the lowest number of genes that are essential for maintaining cell viability and functionality. The bottom-up approach starts from scratch by using biological and nonbiological molecules to build up a “living” artificial organelle or cell [95].

Several researchers have tried to mimic the natural cell or tissue function and reduce the gap between normal and artificial cells. In studies involving artificial cells, microfluidics provides a powerful tool to produce a large number of compartments with different size ranges [96,97]. For instance, a circular design PDMS microfluidic compartmentalized co-culture platform was developed by Park et al. In the fabricated model, neurons and oligodendrocytes are co-cultured in two separate compartments connected by arrays of shallow axon-guiding microfluidic channels. The chip design offers physical and fluidic isolation between the soma and the axon/glia compartments [98]. In an attempt to mimic the structure of biological cells, alginate was used as a biomaterial in artificial systems, and four types of glass microfluidics with flow-focusing or co-flowing droplet generators were used to produce alginate droplets. The generated alginate microgels exhibited various architectures, including individual monodisperse or polydisperse beads, small clusters, and multicompartment systems [99]. For cell culture, microfluidic systems are mainly fabricated with silicon, PDMS, and borosilicate. These materials have been used to test the mammalian embryos within microfluidic systems [100]. Moreover, many microfluidic devices have been reported to enhance cell growth, differentiation, and micro-environmental changes in various perfusion system [101,102,103,104]. In 3D cellular environment, combining PDMS and hydrogel into hybrid device has been used to produce 3D-ECM of aligned for endothelial cell cultures [105]. A study by Leclerc et al. illustrated that the culture of fetal human hepatocytes (FHHs) microfluidic bioreactors is promising for liver tissue engineering. They found that the albumin production by FHHS was four times higher than in static culture which can be influenced to the potentiality of fetal liver cells maturation [106].

These fabricated models show the use of a variety of polymers as distinctive biomaterials and the ease of using microfluidic platforms, which can be used to construct simple mimics of cellular environments or cellular architectures, and thus offer a promising approach for synthesizing bioarchitectures. Table 3 summarizes some studies used polymers and microfluidics in applications described in this review.

## 6. Conclusions

The combination of natural/synthetic polymers and new biofabrication techniques, such as microfluidics, offers promising approach for tissue engineering scaffolds. Polymers and microfluidics enable rapid prototyping, reliability, as well as easy and low-cost manufacturing in research laboratories and for commercialization. Currently, there are many polymer-based drug delivery systems approved by FDA that are available on the market. Further, for commercial mass production, thermoplastics are used to develop standard microfluidic devices. However, despite scientific progress in biofabrication technologies, we are still in the early stages of the development of microfluidic technology for tissue engineering applications. There are serious obstacles to be overcome in producing a functional, complex, and large-scale system.

## Figures and Tables

**Figure 1 polymers-14-05132-f001:**
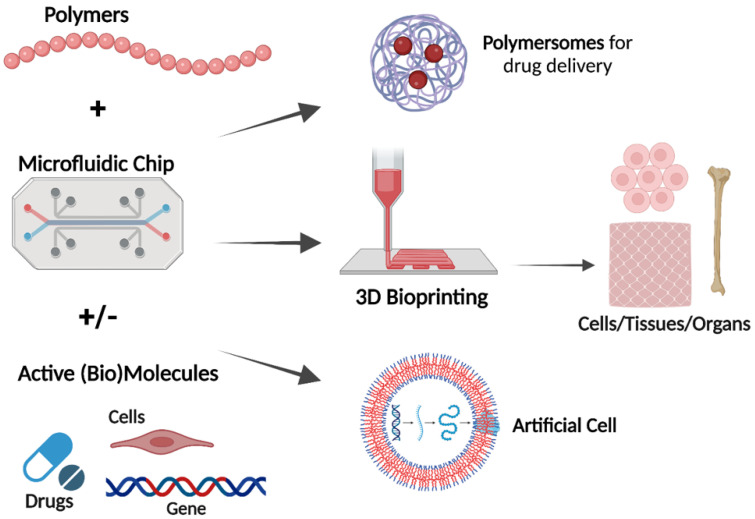
Utilizing of microfluidic chips and polymers for various applications (Created with Biorender.com, accessed on 12 October 2022).

**Figure 2 polymers-14-05132-f002:**
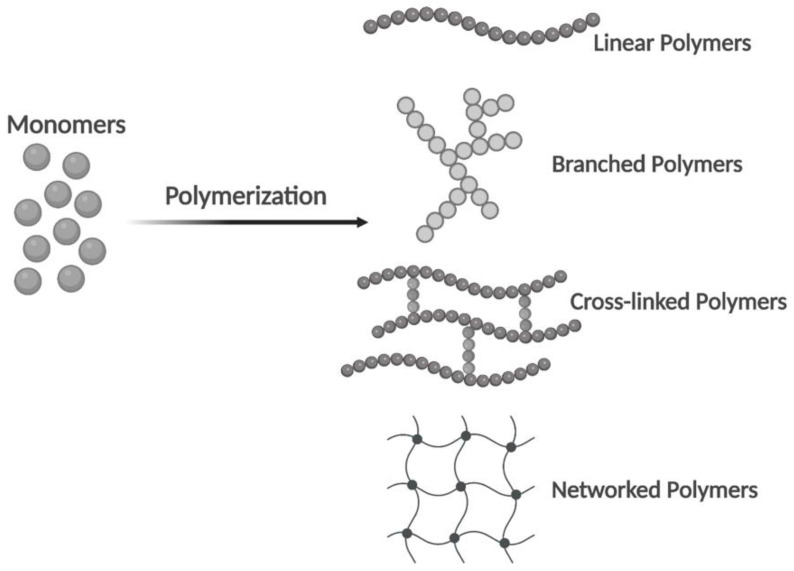
From monomers to polymers. Linear, branched, cross-linked, and networked structures in polymers (Created with Biorender.com, accessed on 12 October 2022).

**Figure 3 polymers-14-05132-f003:**
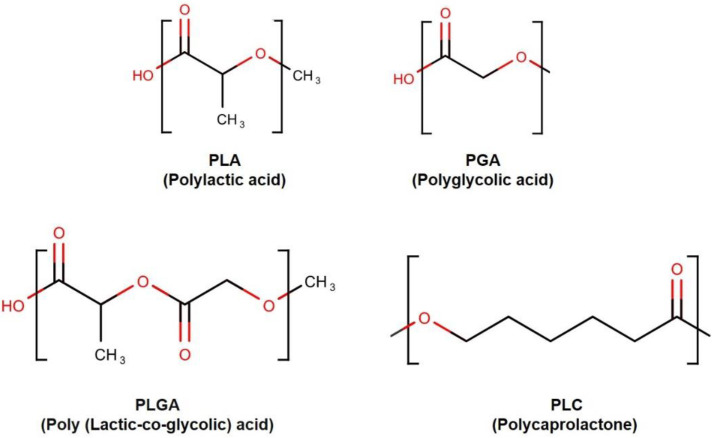
Synthetic bio-degradable polymer structures.

**Figure 4 polymers-14-05132-f004:**
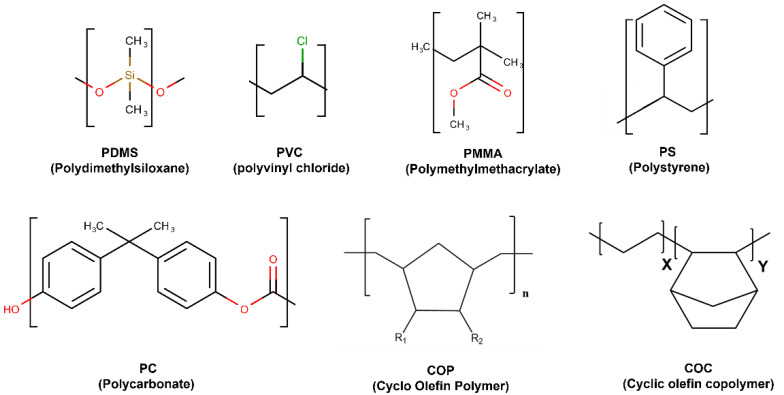
Most used thermostable polymers structures for microfluidic chips.

**Figure 5 polymers-14-05132-f005:**
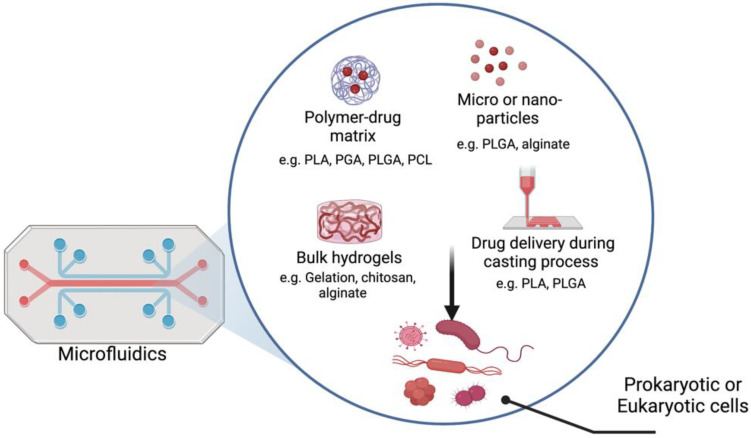
A scheme of different approaches of using polymer for drug delivery. Microfluidics control synthesis of various drug delivery systems. Subsequently, microfluidic chips can be used for cell culture and drug toxicity screening (Created with Biorender.com, accessed on 12 October 2022).

**Figure 6 polymers-14-05132-f006:**
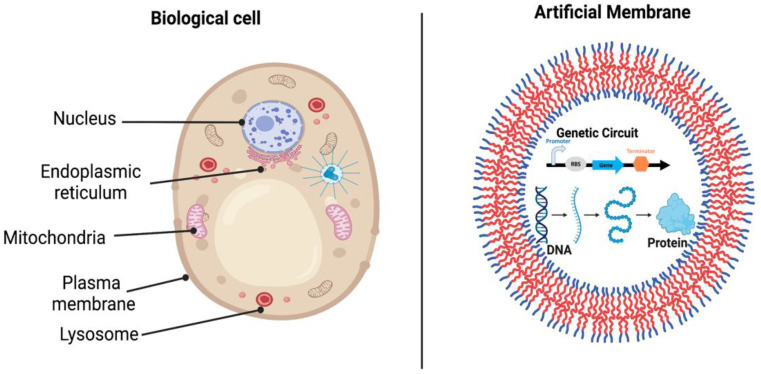
Architecture of typical biological cell and artificial cell. Left: Eukaryotic cell containing different types of organelles. Right: artificial “synthase” cell that mimic that structure of biological cell (Created with Biorender.com, accessed on 12 October 2022).

**Table 1 polymers-14-05132-t001:** Properties of the most used biocompatible thermoplastics in the microfluidic field.

Thermoplastics	Young’s Modulus (Gpa)	Tg (°C)	Tm (°C)	Solubility Parameter δ (MPa)^1/2^	Water Adsorption (%)	O_2_ Permeability (×10^−13^ cm^3^ .cm cm ^−2^ s^−1^ Pa^−1^)	Transparency	Auto-fluorescence	Study
**Polymethylmethacrylate** **(PMMA)**	2.4–3.4	105	250–260	20.1	0.1–0.4	0.1	Transparent	Low	[42]
**Polyethylene terephthalate** **(PET)**	2–2.7	70	255	20.5	0.16	0.03	Transparent	Medium	[43]
**Polypropylene** **(PP)**	1.5–2	−20	160	16.3	0.01–0.1	1.7	Both opaque and transparent	Medium	[41,44]
**Polystyrene** **(PS)**	3–3.5	95	240	18.7	0.02–0.15	2	Transparent	High	[45]
**Polycarbonate** **(PC)**	2.6	145	260–270	19.4	0.23	1	Transparent	High	[41]
**Polyvinyl chloride** **(PVC)**	2.4–4.1	80	100–260	19.4	0.04–0.4	0.04	Transparent	High	[46]
**Polyamide** **(Nylon)**	2.5	47–60	190–350	28	1.6–1.9	0.03	Transparent	High	[47]
**Polytetrafluoroethylene** **(PTFE)**	0.4	115	326	12.6	0.005–0.01	3	Transparent	High	[48]
**Polyetheretherketone (PEEK)**	4–24	143	343	21.9	0.1–0.5	0.1	Opaque	N/A	[49]

**Table 2 polymers-14-05132-t002:** Some studies using polymers for microfluidics devices for various biological applications.

Polymer	Cell Type	Application	Study
Polydimethylsiloxane (PDMS)	Alveolar epithelial cells, Macrophages, *Mycobacterium tuberculosis*	Rapid and uncontrolled bacterial growth in the mammalian cells can cause a surfactant deficiency in the lung-on-chip infection model.	[52]
PDMS, Carboxymethylated cellulose nanofibrils (CNF)	HCT 116 colon cancer cell	The functionalized chip was able to capture the cancer cells from the whole blood with >97% efficiency which may use as rapid diagnostic tool.	[53]
PDMS, Dimethylallylamine (DMAA)	*Escherichia coli*	The encapsulated bacteria with a membrane with a selective permeability of tetracycline cultured on the PDMS composition and functionalized with DMAA inhibit the bacterial growth which can be used as a diagnostic tool to evaluate the bacterial resistance.	[54]
PDMS microchannel layer and PDMS membrane	Human mesenchymal stem cells (hMSCs)	The two layer-microfluidic chips with three different stretching modes (uniaxial, radial, and gradient) showed different cell responses which may enhance the study of cells on biomaterials under various stretching stimuli.	[55]
Combination between PDMS and polymer substrate using a PrimeCoat-Epoxy adhesive layer by selective stamp bonding	Human lung epithelial cells	The cells cultured inside the device showed a similar viability comparing to the conventional cell culture technique.	[56]
Rapidly Integrated Debubbler (RID) from PMMA	human umbilical vein endothelial cells	The RID module showed a potential method to prevent the bubble entry into the microfluidics which may lead to device delamination and cell damage.	[57]
PDMS champers separated by thin layer of polyester (PE) membrane	Primary human small airway epithelial cells	The microfluidics airway system showed a highly controllable and readily accessible physiologic pulmonary environments tailored for lung epithelial cells.	[58]
Combination of PDMS hydrophilic surface treatment and vacuum filling system equipped with bubble trap.	Mouse pancreatic islets	The system showed normal cell viability and morphology, normal insulin secretion, and normal intracellular calcium signaling.	[59]
PDMS	Endothelial cells	The actin filaments alignments directions of the cells cultured in microfluidics channels was significantly higher compering to the cells cultured in the static condition.	[60]
PDMS-glass	Human umbilical vein endothelial cells (VECs)	The synergistic effect of wall shear stress (WSS) and adenosinetriphosphate (ATP) signals played a vital role in the VEC Ca2+ signal transduction on the microfluidic device.	[61]
Photopolymer and chitosan	Hepatic oval cells (HOCs)	Electrochemical sensor is developed to rare cancer cells. Photopolymer is used to construct a 3D-printed continuous flow system and a chitosan film is served as a scaffold for the immobilization of anti-OV6-antibodies.	[62]

**Table 3 polymers-14-05132-t003:** Summary of some studies used polymers and microfluidics for several applications include drug delivery, 3D printing, and artificial cells.

Polymers	Microfluidic Chip Type	Applications	Study
Alginate	Alginate-based bioinks with cartilage cells used to print hollow constructs	The vessels-like printable microfluidic channels were capable to transport nutrients, biomolecules, oxygen through the construct and can support cell growth.	[13]
PLGA	Quartz Droplet X-Junction Chip	insulin was encapsulated into PLGA nanoparticles and then appended with heparin sulfate for oral insulin delivery.	[75]
PDMS	Deep channels or single layer pattern using soft lithography method	High-throughput drug screening can perform using a single chip where enzymatic assays are in picolitre-scale droplets.	[107,108]
PLGA	3D flow-focusing microfluidic chip	Indomethacin was encapsulated into microparticles to develop in silico tool to predict size particles.	[77]
Collagen and alginate	PDMS microfluidic encapsulation device	3D microenvironment of human tumor developed by encapsulating MCF-7 cancer cells in the collagen core of microcapsules with an alginate hydrogel shell for miniaturized 3D culture. Then the cytotoxicity of doxorubicin hydrochloride was assessed.	[109]
Chitosan and alginate	Coaxial flow microfluidic chip	HepG2 cells encapsulated in the chitosan-alginate fibers to guide growth, alignment, and migration of encapsulated cells.	[110]
PDMS and graphene oxide (GO)	Nano-sized GO -modified nanopillars on microgroove hybrid polymer array (NMPA) were fabricated using sequential laser interference lithography and microcontact printing technique.	Mouse myoblast cells (C2C12) were significantly differentiated into skeletal muscle cells on the micro-sized line pattern with GO coating (<10 nm).	[111]
1,6-hexanediol diacrylate (HDDA), lauryl acrylate (LA), polyethylene glycol diacrylate (PEGDA)	3D-printed microfluidic droplet generator	the hydrophobic HDDA/LA 3D printing resin allows droplet formation in 3D-printed planar microfluidics for the basic geometries while hydrophilic PEGDA resin allows droplet formation in non-planar 3D geometry.	[87]
Elastin-like protein (ELP)	Two custom-designed chips: one with ready-made channel and another with sacrificial gel-made channel	ELP hydrogels with cell-adhesive RGD amino acid sequence was used as bioinks for constructing 3D in vitro models with on-chip vascular-like channels. The developed model was compatible with both single cell suspensions of neural progenitor cells (NPCs) and spheroid aggregates of breast cancer cells.	[112]
PDMS	Microfluidic-based droplet system	The high-throughput tree-branched microfluidic droplet system for multicellular spheroids formation showed a high protentional to mimic the in vivo solid tumor structure with heterogeneous cell types and for anti-cancer drug screening applications.	[113]
Alginate	Flow-focusing or co-flowing droplet generators	Generated alginate microgels exhibited various architectures, including individual monodisperse or polydisperse beads, small clusters, and multicompartment systems.	[99]
Au-PEG-PFPE diblock-copolymer surfactant	Microfluidic flow-focusing junction	Lipid vesicles (LUVs or GUVs) were encapsulated into copolymer-stabilized droplets. Generated synthetic cells were able to be loaded with biomolecules, such as FoF1-ATP synthase.	[114]

## Data Availability

Not applicable.

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
