# Peer review of "Role of Polymers in Microfluidic Devices"

_polymers, 2022, doi:10.3390/polym14235132_

Round 1
Reviewer 1 Report
In this manuscript, Damiati et al reviewed recent advances of polymer based microfluidic devices for biomedical applications and polymers for drug delivery, 3D printing and artificial cells. Overall, the development and employment of polymer for microfluidic devices is an interesting topic and the manuscript is in a good structure. Below are my comments:
1. Lines 40-44: ‘Research manuscripts reporting large datasets…’ seems to be irrelevant to the manuscript and needs to be deleted.
2. The usage of the term ‘biopolymer’ in this manuscript is a little confusing. Biopolymer is typically considered as natural polymers produced by living organisms, but in this manuscript PDMS and thermoplastics are classified as biopolymers. I suggest use a different term like ‘biomedical polymer’ instead of ‘biopolymer’ to avoid confusion.
3. Figures 3&4 are in poor quality. Table 1 may be edited to have smaller font size to avoid too many lines in a single cell.
4. Figure 5: what’s the purpose of displaying prokaryotic/eukaryotic cells? Please add more explanations about how they are related to drug delivery.
Minor:
1. Line 56: the limited availability ‘of’ surface modification
2. Lines 71-73: ‘Synthetic polymers are…’ usage of multiple ‘and’ is grammatically incorrect.
3. All latin phrases such as ‘in vivo’, ‘in vitro’, ‘in situ’ should italic.
4. Line 129: ‘water stability’ is vague. Does that mean structure stability(rigidity) or molecule stability to resist hydrolysis?
5. Line 230: ‘solvents such alcohol’ missing an ‘as’
6. Line 277: ‘onto’ should be ‘into’
7. Line 281: ‘that necessitate following the changes..’ is confusing. Please rephrase to make it clear.
8. Lines 341-343: ‘Although 3D printing…’ is not a complete sentence.
9. Figure 6 is misrepresented as ‘Figure 3’
Author Response
Many thanks for the respected reviewer for her/his valuable comments that helped us to improve the manuscript. The suggested additional information has been added to the revised manuscript.
- Lines 40-44: ‘Research manuscripts reporting large datasets…’ seems to be irrelevant to the manuscript and needs to be deleted.
Thanks for pointing this to us. Irrelevant section has been deleted.
- The usage of the term ‘biopolymer’ in this manuscript is a little confusing. Biopolymer is typically considered as natural polymers produced by living organisms, but in this manuscript PDMS and thermoplastics are classified as biopolymers. I suggest use a different term like ‘biomedical polymer’ instead of ‘biopolymer’ to avoid confusion.
The term has been corrected in the revised version of the manuscript.
- Figures 3&4 are in poor quality. Table 1 may be edited to have smaller font size to avoid too many lines in a single cell.
Figures and tables have been modified.
- Figure 5: what’s the purpose of displaying prokaryotic/eukaryotic cells? Please add more explanations about how they are related to drug delivery.
Caption of Fig. 5 has been modified.
‘Figure 5. A scheme of different approaches of using polymer for drug delivery. Microfluidics control synthesis of various drug delivery systems. Subsequently, microfluidic chips can be used for cell culture and drug toxicity screening.’
Minor:
All minor comments have been taken into consideration and the manuscript edited professionally.
- Line 56: the limited availability ‘of’ surace modification
It has been addressed and modified.
- Lines 71-73: ‘Synthetic polymers are…’ usage of multiple ‘and’ is grammatically incorrect.
It has been addressed and modified.
- All latin phrases such as ‘in vivo’, ‘in vitro’, ‘in situ’ should italic.
They have been addressed and modified.
- Line 129: ‘water stability’ is vague. Does that mean structure stability(rigidity) or molecule stability to resist hydrolysis?
This point has been modified.
‘Whereas self-assembled β-sheet structures are responsible for the mechanical stability, and water insolubility of SF, the amorphous regions,….’
- Line 230: ‘solvents such alcohol’ missing an ‘as’
It has been addressed and modified.
- Line 277: ‘onto’ should be ‘into’
It has been addressed and modified.
- Line 281: ‘that necessitate following the changes..’ is confusing. Please rephrase to make it clear.
‘are less suitable for applications that require further reactions inside the microfluidic devices under a microscope’
- Lines 341-343: ‘Although 3D printing…’ is not a complete sentence.
This point has been corrected.
- Figure 6 is misrepresented as ‘Figure 3’
This point has been corrected.
Reviewer 2 Report
The authors review current biostable and biodegradable polymers for biomedical applications. I appreciate the authors’ efforts to put all of these achievements into a review paper. Before considering publication, I have some suggestions for the completeness of this review.
1. The review content actually covers border topics than microfluidic systems. I would suggest changing the review title to fit the content better. Figure 1 is a good summary, and it can help to pick a good review title.
2. Please check the Introduction section carefully. I think the authors’ editing comments got accidentally embedded in the paragraphs. Please check the descriptions after reference [2] and between paragraphs one and two.
3. Figure 1 is a very good summary of the polymers been applied in this field. However, the introduction failed to provide a good overview using this figure. Please provide an Introduction that can give readers a big picture of polymers being applied in biomedical applications, particularly on micro- and nano-scale. The role of microfluidics should also be clear.
4. The authors provide a good Table to summarize the applications of biostable polymers. Please consider the same for biodegradable polymers, drug carriers, and 3-D printing.
5. For the biostable polymers, I think the authors got COP and COC left out. Please consider including them in the review.
Author Response
Many thanks for the respected reviewer for her/his valuable comments that helped us to improve the manuscript. The suggested additional information has been added to the revised manuscript.
- The review content actually covers border topics than microfluidic systems. I would suggest changing the review title to fit the content better. Figure 1 is a good summary, and it can help to pick a good review title.
The title has been changed to ‘Role of Polymers in Microfluidic Devices’.
- Please check the Introduction section carefully. I think the authors’ editing comments got accidentally embedded in the paragraphs. Please check the descriptions after reference [2] and between paragraphs one and two.
Thanks for pointing this to us. Irrelevant section has been deleted.
- Figure 1 is a very good summary of the polymers been applied in this field. However, the introduction failed to provide a good overview using this figure. Please provide an Introduction that can give readers a big picture of polymers being applied in biomedical applications, particularly on micro- and nano-scale. The role of microfluidics should also be clear.
The introduction has been modified and additional published articles have been cited.
- The authors provide a good Table to summarize the applications of biostable polymers. Please consider the same for biodegradable polymers, drug carriers, and 3-D printing.
A new table has prepared.
- For the biostable polymers, I think the authors got COP and COC left out. Please consider including them in the review.
The two polymers are considered in the revised version of the manuscript and in Fig.4.
Reviewer 3 Report
The review article in general is not organized considering the title and the section headings and many current research works are missing because of that. The article needs to be organized and streamlined properly in meaningfully named sections with relevant figures. There are major flaws that need to be addressed.
Some issues with this review article are mentioned below.
1. The title of this article “Polymers-based microfluidic devices” in general means the use of polymers as a constituent in building microfluidic devices which is understandable but the headings for different sections such as “Polymers as Drug carriers” or “Polymers as bioinks for 3D printing” is adding confusion and breaking the flow of the review article. Are you referring to the drug carriers as an integral part of the microfluidic device?
2. The article title needs to be changed to “Role of polymers in microfluidic devices”. In section 2, the heading should have been “Polymers used in microfluidic devices”. The word “for fabrication” makes it look as if the microfluidic device itself is made out of these polymers such as PDMS, PMMA, PS, PVC, hydrogels, etc because these materials can serve as the building chip materials for microfluidic devices and not just some parts of microfluidic devices as mentioned in the review. The authors have completely confused the microfluidic device fabrication base materials with additional components used in modifying the base microfluidic devices in some cases.
A. Read reference 35 in detail by Ren et. al. mentioned in the article for a better understanding of materials used as device building blocks.
B. Please read the section “Microfluidics-based in vitro models” from the article “Advances in modeling Alzheimer’s disease in vitro”, DOI: 10.1002/anbr.202100097 as an example of how hydrogels acted as the base material for the microfluidic device itself. The same hydrogels can be used just to form channels on top of an already-made microfluidic platform as well.
3. The article has no figures showing an existing microfluidic device that is already in use by other researchers. The addition of images showing existing microfluidic devices in every section will add more interest when being referred to in the main contextual sections.
4. Figures are not that clear in this review, e.g. Figure 5- Polymers as drug carriers, is not clear. If they are of different types, they need to be sub-sectioned in the contextual area with individual examples of each type. The section is looking like the information is provided as a cluster.
Author Response
Many thanks for the respected reviewer for her/his valuable comments that helped us to improve the manuscript. The suggested additional information has been added to the revised manuscript.
- The title of this article “Polymers-based microfluidic devices” in general means the use of polymers as a constituent in building microfluidic devices which is understandable but the headings for different sections such as “Polymers as Drug carriers” or “Polymers as bioinks for 3D printing” is adding confusion and breaking the flow of the review article. Are you referring to the drug carriers as an integral part of the microfluidic device?
The title and subtitles have been modified, and the manuscript has been carefully reviewed.
- The article title needs to be changed to “Role of polymers in microfluidic devices”. In section 2, the heading should have been “Polymers used in microfluidic devices”. The word “for fabrication” makes it look as if the microfluidic device itself is made out of these polymers such as PDMS, PMMA, PS, PVC, hydrogels, etc because these materials can serve as the building chip materials for microfluidic devices and not just some parts of microfluidic devices as mentioned in the review. The authors have completely confused the microfluidic device fabrication base materials with additional components used in modifying the base microfluidic devices in some cases.
Thanks for the suggested title. The title has been changed.
- Read reference 35 in detail by Ren et. al. mentioned in the article for a better understanding of materials used as device building blocks.
The article has been read.
- Please read the section “Microfluidics-based in vitro models” from the article “Advances in modeling Alzheimer’s disease in vitro”, DOI: 10.1002/anbr.202100097 as an example of how hydrogels acted as the base material for the microfluidic device itself. The same hydrogels can be used just to form channels on top of an already-made microfluidic platform as well.
Thanks for sharing this interesting article. It has been read.
- The article has no figures showing an existing microfluidic device that is already in use by other researchers. The addition of images showing existing microfluidic devices in every section will add more interest when being referred to in the main contextual sections.
Figures and their captions have been carefully reviewed. For the sake of simplicity, we prepared our figures to provide an overview of basic information.
- Figures are not that clear in this review, e.g. Figure 5- Polymers as drug carriers, is not clear. If they are of different types, they need to be sub-sectioned in the contextual area with individual examples of each type. The section is looking like the information is provided as a cluster.
More information were added into the section and Figure 5 has been modified.
Round 2
Reviewer 1 Report
The authors addressed all my previous questions, but I think the introduction part can be further improved to be cohesive and clear. For example, some of the sentences are not logically connected. I would suggest authors to further polish the introduction before publication.
Author Response
The authors addressed all my previous questions, but I think the introduction part can be further improved to be cohesive and clear. For example, some of the sentences are not logically connected. I would suggest authors to further polish the introduction before publication.
We want to thank the reviewer for his/her comment. As suggested by the reviewer, we have carefully revised and modified the manuscript.
Reviewer 2 Report
The authors replied to my comments with an updated revision. Most of the comments are properly replied. However, the authors still failed to provide a broader view in the Introduction section using figure 1. The current Introduction section does not show a direct linkage to figure 1 and the contents of the following review sections. I strongly suggest the authors revised the introduction to highlight the review contents in the Introduction section.
Author Response
The authors replied to my comments with an updated revision. Most of the comments are properly replied. However, the authors still failed to provide a broader view in the Introduction section using figure 1. The current Introduction section does not show a direct linkage to figure 1 and the contents of the following review sections. I strongly suggest the authors revised the introduction to highlight the review contents in the Introduction section.
We want to thank the reviewer for his/her comment. As suggested by the reviewer, we have carefully revised and modified the manuscript.
Reviewer 3 Report
The manuscript could have been made better if the authors would have added few more references in each section mentioning more recent studies. For a mini review also, the number of mentioned studies is not adequate.
Author Response
The manuscript could have been made better if the authors would have added few more references in each section mentioning more recent studies. For a mini review also, the number of mentioned studies is not adequate.
We want to thank the reviewer for his/her comment. As suggested by the reviewer, we have carefully revised and modified the manuscript.